# Tetragonal–Cubic Phase Transition and Low-Field Dielectric Properties of CH_3_NH_3_PbI_3_ Crystals

**DOI:** 10.3390/ma14154215

**Published:** 2021-07-28

**Authors:** Roxana E. Patru, Hamidreza Khassaf, Iuliana Pasuk, Mihaela Botea, Lucian Trupina, Constantin-Paul Ganea, Lucian Pintilie, Ioana Pintilie

**Affiliations:** 1National Institute of Materials Physics, Atomistilor 405A, 077125 Magurele, Romania; roxana.patru@infim.ro (R.E.P.); iuliana.pasuk@infim.ro (I.P.); botea.mihaela@infim.ro (M.B.); Lucian.Trupina@infim.ro (L.T.); paul.ganea@infim.ro (C.-P.G.); pintilie@infim.ro (L.P.); 2Department of Materials Science and Engineering, University of Connecticut, 97 North Eagleville Road, Storrs, CT 06269, USA; hamid.khassaf@gmail.com; 3Institute of Materials Science, University of Connecticut, 97 North Eagleville Road, Storrs, CT 06269, USA

**Keywords:** MAPI, ferroelectricity, phase transition, conductivity, relaxation processes

## Abstract

The frequency and temperature dependence of dielectric properties of CH_3_NH_3_PbI_3_ (MAPI) crystals have been studied and analyzed in connection with temperature-dependent structural studies. The obtained results bring arguments for the existence of ferroelectricity and aim to complete the current knowledge on the thermally activated conduction mechanisms, in dark equilibrium and in the presence of a small external a.c. electric field. The study correlates the frequency-dispersive dielectric spectra with the conduction mechanisms and their relaxation processes, as well as with the different transport regimes indicated by the Nyquist plots. The different energy barriers revealed by the impedance spectroscopy highlight the dominant transport mechanisms in different frequency and temperature ranges, being associated with the bulk of the grains, their boundaries, and/or the electrodes’ interfaces.

## 1. Introduction

Current developments for clean energy production and the continuous effort to improve the photovoltaic-based devices have brought attention to metalorganic lead–halide perovskite materials. These types of compounds are used in perovskite solar cells (PSCs), a new generation of potentially low costs solar cells with power conversion efficiencies (PCE) exceeding 24% [1,2,3,4,5,6].

Despite the tremendous rapid increase in the value of PCE, there are still issues to be resolved in order to enhance the stability and the electrical performance of PSCs. For example, the piezoelectric and electrostrictive performances of methylammonium lead iodide (CH_3_NH_3_PbI_3_—MAPI) single crystals depend on the strain induced by the applied electric fields. When it comes to PSCs, the electrostrictive strain between the perovskite grains impacts their stability/degradation under illumination [7]. Another topic of interest is the presence of bulk ferroelectricity at room temperature (RT), a property that could explain why these materials are so efficient in converting light into electricity. Ferroelectricity may exist in MAPI because of the spontaneous and persistent tilting, distortions, and rotation of the octahedral PbI_6_. The formation of polarized domain structures is favored by the permanent molecular dipoles existing in MAPI. The electric and elastic dipoles associated with the highly rotationally CH_3_NH_3_ (MA) molecules are reorienting in correlation with the tilt configurations of the PbI_6_ octahedra, which, in turn, generates an overall electric dipole between MA cation and PbI_6_ anion sites lattice [8,9,10].

The change in crystalline symmetry that occurs with varying the temperature is due to phase transitions. The possible structural configurations of perovskites have been deduced and classified by Glazer [11]. The ferroelectric ordering in rhombohedral, orthorhombic, and tetragonal phases is determined by the noncentrosymmetric unit cell with a single heavy metal cation atom. However, in MAPI, the MA molecular ion is rotationally mobile at room temperature, and it is argued that the ferroelectricity in MAPI is probably related to the dynamics of MA. Most of the theoretical works calculated the spontaneous polarization to be in the 4–14 µC/cm^2^ range [12,13,14,15,16], of much lower values, compared with inorganic perovskites. Additionally, several piezoelectric force microscopy (PFM) studies claimed to provide evidence for ferroelectricity [17,18], while in other studies, a persisting ferroelectric polarization has not been observed [19,20]. In the study performed by Cordero et al. [21], it is argued that in MAPI, the reorientation dynamics of the electric and elastic dipoles associated with the MA cation is competing with the anti-Ferro distortive modes originating in the PbI_6_ in-phase tilting, thus preventing the ordering into a ferroelectric phase. On the other hand, previous studies have shown that MAPI undergoes several structural phase transitions from an orthorhombic to a tetragonal structure at −113 °C and to the paraelectric cubic phase over 54 °C [22,23], resulting in randomly oriented ferroelectric domains in the proximity of the room temperature.

Bulk polarization may appear even in the cubic phase (in the absence of ferroelectric effect) when the polar MA molecule reduces its lattice symmetry from I4/mcm to noncentrosymmetric I4cm, by changing the alignment with the center of the perovskite structure [14]. There are several related forms of polarization resulted from the displacement of Pb within the PbI_6_ octahedral, originated from the orientation or from the shift of MA^+^ dipole decentering relative to the negative charge center of the PbI_3_^−^ [20]. Previous studies suggested that the migration mechanism of MA^+^ cations and I^−^ anions and their accumulation at the grain boundaries/interfaces involve ionic mobility within distinctive thermally activated processes [24], are dependent on the measurement conditions and protocols [25,26], and significantly contribute to the charge transport in MAPI thin films.

In this study, the structural, electrical, ferroelectric, dielectric, and relaxation properties of MAPI crystals are investigated. The aim is to understand the electrical behavior of hybrid perovskite crystals and establish the connection with the nature of the migrating species and their impact on the dielectric permittivity, losses, and electrical conduction in the temperature range of interest for PSCs of −30 °C–110 °C. In addition, the performed impedance measurements bring arguments for the existence of ferroelectricity in MAPI crystals.

## 2. Materials and Methods

Methylammonium lead iodide (CH_3_NH_3_PbI_3_, MAPI) single crystals were synthesized using inverse temperature crystallization (ITC) technique in which, the organo–lead trihalide perovskites materials exhibit inverse solubility [27]. One molar MAPI solution was prepared by dissolving PbI_2_ (Sigma-Aldrich 99.9995%, St. Louis, USA) and MAI (Dyesol) in stoichiometric (1:1) ratio in γ–butyrolactone (GBL) solvent. The precursors were added to GBL after the solvent was heated up to 60 °C. After a continuous mixture of the precursors, the solution was filtered using a polytetrafluoroethylene (PTFE) filter with 0.2 µm pore size. Two milliliters of the filtrate were then placed in a vial and introduced in a silicon oil bath at 115 °C. The setup was kept extremely stable and undisturbed to prevent fluctuations that would result in the formation of several tiny crystals. In this way, single crystals large enough for electrical characterization were obtained. The procedure was carried out under ambient conditions and humidity of ~40%. It took about 4–5 h to form 3–5 mm size crystals. The crystals were then fished out, immediately dried with tissue, and put in a vacuumed desiccator to prevent exposure to humidity.

Many common metallic electrodes strongly react with MAPI perovskite [28]. Ti/Au top and bottom electrodes have been chosen for making good electrical contacts and reducing the hole traps at the interfaces [29]. The Ti/Au layers, with an area of ~7 mm^2^ and thickness of 100 nm, were deposited by rf-magnetron sputtering.

The crystal structure was analyzed by X-ray diffraction (XRD) using a Rigaku-SmartLab diffractometer (Rigaku Corporation, Tokyo, Japan) with DHS 1100 temperature chamber (Anton Paar GmbH, Austria), equipped with a copper anode X-ray tube operated at 40 kV and 40 mA, and a HyPix 3000 detector used in 1D mode. In order to avoid the influence of the preferred orientation of the crystal, the XRD analyses have been performed on a powder sample obtained by crashing and milling a crystal. A zero-background silicon sample holder was used for hosting the powder. In situ XRD patterns were recorded at different temperatures between 30 °C and 110 °C, with a ramp of 10 °C/min and 2 min’ stabilization before starting each measurement. The scans were performed in Bragg–Brentano geometry, in the angular range 2θ = 11.5°–33°, step size 0.01°, with a scan speed of 1 deg/min. The structure parameters were determined by fitting the whole powder pattern using the Bruker-TOPAS v.3 program (Bruker AXS Inc., Madison, WI, USA) in the fundamental parameters approach. The surface morphology was investigated by using a Gemini 500 scanning electron microscope (SEM) from Zeiss and an MFP 3D SA atomic force microscope (AFM) from Asylum Research equipped with a PFM. In order to enhance the PFM sensitivity, the images were acquired using an Olympus AC240-TM cantilevers (l = 240 µm, spring constant = 2 N/m, Pt coated) operated near resonance frequency in the single frequency mode. Electrical measurements were performed using an Alpha-A Novocontrol dielectric spectrometer meter system allowing the variation of temperature while maintaining the sample in a controlled nitrogen atmosphere. The measurements were performed first during cooling from 110 °C to −30 °C and then back during heating to 110 °C, in steps of 5 °C, using an a.c. voltage of 0.1 V and sweeping the frequency from 10^−2^ Hz to 10^7^ Hz.

## 3. Results and Discussion

### 3.1. Structural and Microstructural Investigation

The XRD investigations were performed at the following temperatures: 27 (RT), 30, 50, 55, 60, 65, 70, 90, and 110 °C. The XRD patterns corresponding to the two extreme temperatures in our study, RT of 27 °C and 110 °C are presented in Figure 1a. The phase identification and line indexing are based on the ICDD Database Release 2020.

At 27 °C, the structure identifies with tetragonal MAPbI_2_, space group I4/mcm (according to ICDD-PDF4 # 01-085-5508), while at 110 °C the structure is cubic, space group Pm-3m (according to ICDD-PDF4 # 00-069-0999). As the ICDD-2020 database does not contain MAPbI_2_ with primitive cubic structure, the pattern at 110 °C was compared with formamidinium lead iodide with cubic Pm–3m structure, which resembles MAPbI_2_ except the organic group, CH_5_N_2_ instead of CH_6_N. The patterns also show a very small content of PbI_2_. The X-ray diffractograms corresponding to all the considered temperatures are presented in Figure 1b (zoomed views of the significant regions). The most obvious observed feature is the monotonous decrease of the tetragonal splitting amplitude with increasing the temperature until the corresponding tetragonal lines in the 13.9°–14.2°, 28.1°–28.5°, and 31.5°–32° ranges of 2θ apparently merge into one between 55 °C and 60 °C, marking the transition to the cubic phase of MAPI. However, the merged peak is slightly asymmetric at 60 °C. On the other hand, the only peaks of the tetragonal structure that are well separated from the cubic ones are 211 at 2θ ≈ 23.5° and 213 at 2θ ≈ 29.9° (superlattice peaks). The relative intensities of these peaks decrease gradually until disappear at about 70 °C. These features suggest the coexistence of the cubic and tetragonal phases (at least) in the temperature range 55–70 °C. Whitfield et al. [30] found that there is a wide range of cubic and tetragonal phase coexistence from 300 to 330 K. According to the work of Kawamura et al. [31], for a single crystal MAPI, the decrease of the superlattice peak intensities with increasing the temperature last until the phase transition occurs at ~58 °C. The authors found that this process is related to the rotation of the PbI_6_ octahedra around the *c* axis, and it is accompanied by the decrease of tetragonality. Baikie et al. [23] show that the 211 superlattice reflection persists up to 75 °C, with a large change of intensity at 57 °C. Another general feature of the temperature-dependent diffractograms is the continuous shift of the peaks toward lower angles as the temperature increases. This shift is due to thermal expansion, as it will be shown further on.

The structural evolution with temperature, lattice constants, crystallite sizes, and the microstrain have been estimated by fitting the diffraction patterns according to the Pawley approach, by using the Bruker-TOPAS v.3 software (Bruker AXS Inc., Madison, U.S.A.). This method works without considering a model of atomic positions in the unit cell, avoiding thus the structural modeling of the complex MAPI unit cell. It simulates the line profiles determining the exact positions and line widths, while it takes into account only the space group and the initial lattice constants. In this way, the Pawley method allows the precise determination of the lattice constants values, average crystallite sizes, and the microstrain. The profile of instrumental function was estimated by using the fundamental parameter approach. The zero error was fixed to zero, and the sample displacement parameter was refined at each temperature. The diffractograms were fitted considering a tetragonal structure up to 70 °C and a cubic one above 60 °C, meaning that in the temperature range of 60–70 °C, the data were fitted either with a tetragonal phase or with a cubic one.

The temperature dependences of the refined structure parameters are presented in Figure 2a–d. They result in a good quality of the fit, as presented in Figure 2e for several temperatures. We will discuss in the following the temperature variations of all the refined parameters used for fitting the diffractograms. In order to make easier the comparison between the parameters of tetragonal (a_t_, c_t_) and cubic MAPI structure (a_c_), in Figure 2a, we represented a_t_’ = a_t_/√2 and c_t_’ = c_t_/2 [31]. As the temperature increases, a_t_’ increases and c_t_’ decreases, until they approach values that are similar with the lattice parameter of the cubic phase, a_c_, at approx. 60 °C. Above this temperature, a_t_’ and c_t_’ are very slowly approaching but do not exactly become equal even at 70 °C. This slow change of a_t_’ and c_t_’ above 60 °C could be attributed to the fitting errors. The structure parameter $\overset{3}{\surd }V_{t}^{\prime }$ plotted in Figure 2a has the following meaning: considering a cubic structure with the same unit cell volume as the tetragonal one, V_c_’ = V_t_’ = a_t_’^2^ c_t_’, one can define a volume equivalent cubic lattice parameter a_c_’ = $\overset{3}{\surd }V_{t}^{\prime }$. If this parameter modifies due to thermal expansion only, its values at two different temperatures should be connected by the equation: a_c_’(T_2_) = a_c_’(T_1_) [1 + α (T_2_ − T_1_)], where α is the linear thermal expansion coefficient. In this way, one can determine α from the slope of the straight line drawn between the points. From the linear fit of the $\overset{3}{\surd }V_{t}^{\prime }$ plot one obtains for the tetragonal phase: α_tetragonal_ = 39.8 × 10^−^^6^ K^−1^. This value is very close to that obtained by applying the same procedure for the cubic MAPI above 60 °C: α_cubic_ = 43.6 × 10^−6^ K^−1^. Applying the same procedure, Whitfield et al. have found α_tetragonal_ = 42.4 × 10^−^^6^ K^−1^, and α_cubic_ = 35 × 10^−^^6^ K^−1^ [30]. The plots indicate that the unit cell volume remains practically unchanged at the phase transition from tetragonal to cubic. Figure 2b,c represents the evolution of the microstructural parameters of the MAPI powder. The crystallite size of a crystal phase measures the average distance for which the same type of atomic ordering and the same orientation is preserved. The microstrain is a parameter of a crystalline phase, which measures the local degree of lattice distortions within the crystallites or the structural inhomogeneity over all crystallites in a certain phase. It is expressed by the dispersion of the interplanar spacing fluctuations over the investigated volume of the sample. It is usually expressed as <ε^2^>^1/2^, where ε = Δd/d, and d are interplanar spacings of the lattice [32]. Both the crystallite size and the microstrain determine the broadening of the XRD line. The separation of these two contributions is based on their different dependence on the diffraction angle. Thus, while the broadening caused by size is proportional to 1/cos θ, the one determined by the microstrain depends on tan θ. The size and microstrain values were calculated by using the complex algorithm provided by the TOPAS program. It is worth noting that the degree of lattice distortion determined by XRD refers to the heavy atoms, Pb and I, since they contribute to the scattering more than C, N, and H. The graphs presented in Figure 2b,c show that both the crystallites of the tetragonal MAPI and the lattice disorder/structural inhomogeneity slightly decrease upon heating. This shrinking process stops between 55 and 60 °C and either the lattice distortion of the tetragonal phase increases abruptly if the majority phase is tetragonal, or the cubic phase has already become dominant, and it starts with a highly distorted lattice. Heating further up to 70 °C, the crystallites remain small, but the crystalline ordering rapidly improves. Above 70 °C the size of cubic phase crystallites increases monotonously with temperature, with the concomitant improvement of the atomic ordering. One would expect that the crystallite size and the microstrain are subjected to a larger uncertainty than that of the lattice parameters since they are related to the peak breadths, the evaluation of which being more affected by peak profile anomalies. However, Figure 1b clearly shows that the peaks are monotonously narrowing, starting from 60 °C to 110 °C, which indicates an increase of crystallites size or an improvement of lattice ordering, or both. The refinement of the structure parameters indicates that the peak narrowing is due to the sharp reduction of lattice distortions between 60 and 70 °C and to the increase of size and ordering above 70 °C. Figure 2d shows the evolution of the displacement of the sample’s surface from the center of the goniometer. This experimental parameter has been refined because of its interference with the structure parameters, especially with the lattice parameters. In the case of temperature-dependent XRD measurements, this could vary due to the expansion or shrinkage of the sample. The steep decrease at about 55 °C indicates the sharp expansion of the sample at that temperature. Above 70 °C the expansion continues in a much slower manner.

One can conclude from the XRD investigations that the tetragonal–cubic transformation takes place gradually, starting sharply between 55 and 60 °C and being completed at approximately 70 °C. The transition is accompanied by a sharp volume expansion of the sample, while the unit cell volume expands monotonously from room temperature to 110 °C. In the transition region, the crystal structure is characterized by a high degree of distortion and/or large structural inhomogeneity. Whitfield et al. reported the coexistence of tetragonal and cubic phases in a temperature range of nearly 30 °C and considered this phenomenon as evidence for a first-order phase transition [30].

### 3.2. Surface Morphology

The images of the surface topography, piezoresponse amplitude, and piezoresponse phase in a MAPI crystal measured over 1.5 × 1.5 µm^2^ area are given in Figure 3a–c, respectively. The PFM phase image (Figure 3c) shows strong and clear contrast indicating the presence of ferroelectric domains at the surface of the sample. However, these domains could not be switched due to the large voltage required for polarization reversal (the crystal thickness was about 3 mm). It is worth mentioning that no topographic changes on the film surfaces nor crosstalk between topography and phase were observed during PFM scanning. Accounting also for other PFM results on thinner MAPI samples (250 nm and 100 nm), reported in [17] and [18], where the polarization switching could be achieved, we interpret the phase variation in Figure 3c as indicating the presence of ferroelectric domains at the surface of the sample.

SEM investigations, presented in Figure 4, show large crystalline domains with visible growth terraces, indicating that the sample is not exactly a single crystal (Figure 4a). The fractured inner surface shows sharp edges at the grain’s extremities and small, irregular particles of nonuniform morphologies formed upon the fracture of the sample. The confluence between the multiple grains joined during growing is visible in Figure 4b. There are no obvious defects inside the crystal observable by SEM. On the contrary, the microstructure reveals dense and compact MAPI crystals. However, the fragile intercrystalline nature of the material makes it extremely sensitive to mechanical shocks.

### 3.3. Dielectric Spectroscopy

The real part of the dielectric permittivity (ε’) recorded in the 10 mHz ÷ 10 MHz frequency range during heating between −30 °C and 110 °C is given as a 3D representation in Figure 5 and is indicating a dispersive dielectric polarization.

Three large regions, associated with different processes can be observed: (i) the low-frequency range, with a sharply increasing permittivity with temperature, associated with the thermally activated Maxwell–Wagner mechanism of electrode polarization and dc-ionic conduction [33,34,35]; (ii) the medium range of frequencies, with a softer variation in permittivity, associated with charges accumulated at the grain boundaries producing a wide dielectric relaxation regardless of temperature [34,35]; (iii) high-frequency range, with a weak variation in ε’, associated with the response coming from the bulk of the crystal [35]. The frequency and temperature dependencies will be discussed further according to different kinds of analyses performed.

#### 3.3.1. Dielectric Function and Curie Weiss Analyses

The real part of dielectric permittivity and the loss tangent (tan δ) as a function of temperature for frequencies between 10 Hz and 10 MHz are given in Figure 6. The dielectric data are presented for two frequency ranges, 10 Hz–1 kHz and 10 kHz–10 MHz. The data for all frequencies are given in Appendix A.

In the low-frequency range, ε’ is displaying a very close to a relaxor behavior, with diffuse ferroelectric to paraelectric phase transition extended over a wide temperature range (see Figure 6a). The increasing trend with the temperature of ε’ is frequency dispersive, without having a thermal hysteresis. However, the configured maximum above 30 °C is shifting toward higher temperatures with increasing frequency. The dielectric losses significantly increase with temperature, reaching maximum values around 100 °C (Figure 6c). A slight decrease of ε’ with increasing the temperature is observed for frequencies above 10 kHz (see Figure 6b). For these frequencies, the weaker variation of ε’ is observed between 60 and 90 °C, along with a slight thermal hysteretic behavior. The dielectric losses are also increasing with temperature; however, the measured values remain low over the entire temperature range for frequencies above 10 kHz (see Figure 6d).

The cubic-to-tetragonal phase transition in MAPI has been reported to occur around 54 °C with a thermal hysteresis of 2–5 degrees between heating and cooling [21,22,23]. Additionally, a coexistence of paraelectric and ferroelectric phases in MAPI crystal, manifested via a temperature range where ε’ remains almost constant, was previously reported [30]. This behavior is attributed to a first-order cubic-to-tetragonal phase transition, very close to the tricritical point. Similarly, one can assume that the plateau observed in the temperature dependence of ε’ in Figure 6b (between 60 and 90 °C) roughly marks the tetragonal to cubic phase transition in MAPI crystal.

Particularly, phase transition between distinct crystal structures is due to the organic cations that have a disordered dynamic at high temperatures and freeze at low temperature, and therefore, the centrosymmetric structure disappears leading to the appearance of a spontaneous polarization [36]. The low-symmetry in hybrid organic–inorganic perovskites structures was investigated by first-principles calculations. Egger et al. [37] observed the relatively weak crystal cohesion and bonding in connection with molecular rotation, octahedral distortions, and ionic diffusion. Considering the disordered character of methylammonium ion to which contributions from the displacive character of the PbX octahedron are enclosed, MAPI undergoes an order-disorder type phase transition [38]. The order−disorder and displacive phase transitions coexist [37] due to the significant coupling between the strongly temperature-dependent rotational dynamics of molecular MA cations [39] and the distortions and rearrangements of PbI_6_ anions [40]. Furthermore, the ionic polarization arising in MAPI from the mobility of ions also points out to a displacive first-order phase transition [30,41,42].

A clear determination of the Curie point, associated with the relative maximum value of the dielectric permittivity, $\varepsilon_{m}^{\prime }$, is not an easy task in such cases, yet there are some mathematical solutions. One way is to study the monotony of the given $\varepsilon^{\prime }\left(T\right)$ function, calculate the roots of the first-order derivative $\frac{d\varepsilon^{\prime }\left(T\right)}{dT}=0$ and find the Curie temperature. In addition, the local minimum in the first derivative indicates the coexistence of phases, referred to in the literature as a polymorphic phase boundary [43,44]. Valuable information can also be extracted from the second-order derivative $\frac{d^{2}\varepsilon^{\prime }\left(T\right)}{dT^{2}}$ [45]. Thus, the changes in the $\varepsilon^{\prime }\left(T\right)$ slope will give peaks in the second derivative. From each of the peaks, the transition temperature of two adjacent phases can be precisely identified if no frequency dispersion exists. Figure 7 shows the first- and the second-order derivatives of $\varepsilon^{\prime }\left(T\right)$ for a few eloquent frequencies above 40 Hz.

Above 10 kHz (Figure 7a), even though the dielectric permittivity is monotonically decreasing with increasing temperature, there are a local minimum, a maximum, and inflection points in $\frac{d\varepsilon^{\prime }\left(T\right)}{dT}$, generating the two peaks in $\frac{d^{2}\varepsilon^{\prime }\left(T\right)}{dT^{2}}$ that suggest two joined phases between ~24 °C and ~60 °C and a full cubic phase starting with ~75 °C. In the 40 Hz ÷ 750 Hz frequency range, $\varepsilon^{\prime }\left(T\right)$ reaches maximum values between 70 °C and 102 °C without any other inflection points or local deviations in $\frac{d^{2}\varepsilon^{\prime }\left(T\right)}{dT^{2}}$ (Figure 7b). The shift of $\varepsilon_{m}^{\prime }$ in $\varepsilon^{\prime }\left(T\right)$ to higher temperatures when frequency increases from 40 Hz to 750 Hz is a usual behavior in ferroelectric solid solutions with diffused phase transition (ferroelectric relaxors).

The diffusive behavior is analyzed based on both, the modified and the generalized Curie–Weiss laws. The results are illustrated in Figure 8a,b for heating and cooling cycles, respectively.

The nonlinear fitting (modified Curie–Weiss approach) has been performed considering [46]
(1)$$\varepsilon^{\prime }\left(T\right)=\frac{\varepsilon_{m}^{\prime }}{1+{\left(\frac{T-T_{m}}{\delta_{\gamma }}\right)}^{\gamma }}$$
in which $\varepsilon_{m}^{\prime }$ is the maximum value of the measured $\varepsilon^{\prime }\left(T\right)$, $T_{m}$ temperature corresponding to $\varepsilon_{m}^{\prime }$, while γ and $\delta_{\gamma }$ parameters are related to the transition character, quantifying the diffuseness degree and the $\varepsilon^{\prime }\left(T\right)$ peak broadening.

The interpolation traced on the nonlinear fitting allows verifying if the dielectric data obeys the modified Curie–Weiss law (see the insets of Figure 8a,b) based on the following relation [47]:(2)$$\frac{1}{\varepsilon^{\prime }\left(T\right)}-\frac{1}{\varepsilon_{m}^{\prime }}=C^{-1}{\left(T-T_{m}\right)}^{\gamma }$$
where *C* is the Curie constant. The diffusivity exponent, present in Equations (1) and (2) reaches values between 1.9 and 2, with small variations between heating and cooling. These values, which are found only between 10 Hz- and 1 kHz, are characteristic of ideal relaxor materials with a complete diffuse phase transition [48].

#### 3.3.2. Dielectric Relaxations

Dipoles and electric charges require time to relax after their polarization, aligning, and moving under an applied a.c. electric field, resulting in dipolar and ionic relaxations. The high values of permittivity especially at low frequencies, along with conductivity contributions, place out of the sight the dielectric relaxations. Even so, by the instrumentality of complex permittivity analysis combined with the dielectric modulus approach, a better description of the frequency dependencies of MAPI crystal properties can be achieved. Thus, by analyzing the complex dielectric function, $\varepsilon^{*}\left(\omega ,T\right)=\varepsilon^{\prime }\left(\omega ,T\right)-i\varepsilon^{\prime \prime }\left(\omega ,T\right)$, where $\varepsilon^{\prime }\left(\omega ,T\right)$ is the dielectric permittivity and $\varepsilon^{\prime \prime }\left(\omega ,T\right)$ is the loss factor, the rotational and translational diffusion of dipoles as well as the interfacial polarization can be evaluated. When all dipoles have the same relaxation time, the dielectric function is given by [48]
(3)$$\varepsilon^{*}\left(\omega \right)=\varepsilon_{\infty }+\frac{\Delta \varepsilon }{1+i\omega \tau },$$
where $\varepsilon_{\infty }$ represents the displacement polarization, described as unrelaxed permittivity on high-frequency limits, $\Delta \varepsilon =\varepsilon_{s}-\varepsilon_{\infty }$ and accounts for the dielectric strength, $\varepsilon_{s}$ represents the static permittivity, ω is the angular frequency and τ=τ0×expEaKT is the Debye relaxation time, where Ea is the activation energy, *T* is the temperature K is the Boltzmann constant and $\tau_{0}$ a preexponential factor.

Any deviation of the peak shape from the classical Debye relaxation of the dipoles [49] implies multiple relaxation times behavior. The very strong dispersion of the dielectric permittivity in such a broad temperature and frequency range is most likely based on series of several Debye-type relaxation processes with relaxation times distributed in particular manners, as most often happens in real materials. The Cole–Cole functions are describing the symmetric broadening of the $\varepsilon^{\prime \prime }$ peak shape [50], while the Cole–Davidson functions the asymmetric ones [51]. The Jonscher function is used when $\varepsilon^{\prime \prime }$ obeys a power-law dependence on frequency [52] and the Kohlrausch–Williams–Watts relation for distribution of exponential decays [53,54]. However, regardless of the approach, it remains difficult to separate processes with similar relaxation times and to explain the dynamics of the polar nanoregions. Havriliak and Negami proposed corrections to the Debye equation (Equation (3)), describing the disordered systems by the empirical formulae [55] as follows:(4)$$\varepsilon^{*}\left(\omega \right)=\varepsilon_{\infty }+\frac{\Delta \varepsilon }{{\left[1+{\left(i\omega \tau \right)}^{\alpha }\right]}^{\beta }}$$
where the asymmetry (α) and broadening (β) parameters, 0<α,β≤1, are introduced to model the dielectric spectra [55,56,57].

Figure 9 illustrates the frequency dependence of the complex permittivity when the temperature raises from −30 °C to 110 °C.

The space charge relaxation observed in the dielectric spectra gives rise to loss peaks shifting toward higher frequencies and to a crossing point for ε′ and ε″ when increasing the temperature. The two exponential factors α and β parameters added to the initial Debye equation are useful to explain such a response. The α parameter, related to crystal disorder, has the maximum 1 value only below 75 °C. Above this temperature, α slightly decreases, indicating an increase of the disorder in the crystal. on the other hand, the β parameter gradually increases with temperature. Thus, the broadness of the dielectric dispersion curve reduces with increasing the temperature, and the system is best described by Cole–Davidson model. The dispersive phenomena in our study are ending when β parameter is reaching the 1 value at 110 °C and the dielectric function starts to be described by the Cole–Cole model. One can conclude that the relaxation phenomena observed in the dielectric function, each having its own relaxing time, evolve with temperature and overlap generating, as it will show later on, a large range for the relaxation times.

Widely used to investigate the electrical transport in materials showing long-range conduction and local dielectric relaxations, is the complex dielectric modulus *M**, calculated based on the complex dielectric permittivity according to [58]
(5)$$M^{*}=\frac{1}{\varepsilon^{*}}=\frac{\varepsilon^{\prime }}{\left[{\left(\varepsilon^{\prime }\right)}^{2}+{\left(\varepsilon^{\prime \prime }\right)}^{2}\right]}+j\frac{\varepsilon^{\prime \prime }}{\left[{\left(\varepsilon^{\prime }\right)}^{2}+{\left(\varepsilon^{\prime \prime }\right)}^{2}\right]}=M^{\prime }+jM”$$

Figure 10 shows the frequency dependence of the real *M*′ and imaginary *M*″ parts of *M** between −30 °C and 110 °C. Two distinct contributions giving rise to shoulders/maxima in *M*′ and *M*″ are observed, and they are labeled in the figure as Regions I and II. Region I extend from 10^−2^ Hz to ~10^3^ Hz and Region II from 10^0^ Hz to ~10^4^ Hz. The involved processes give rise to relaxation phenomena outlining a joint activity from 10^0^ Hz to ~10^3^ Hz associated with effects occurring on the grain boundary and conductive process of electrodes polarization.

The $M^{\prime }\left(\nu \right)$ isotherms in Figure 10a have constant values at high frequencies and tend asymptotically to zero with lowering the frequency, more rapidly as the temperature is increased. At low temperatures, the non-zero values of $M^{\prime }\left(\nu \right)$ in Region I (10^−2^ Hz ÷ 10^3^ Hz) indicate that there are contributions from electrode polarization, which resulted most likely from ions accumulation at the perovskite–electrode interface [59,60]. $M^{\prime }\left(\nu \right)$ tends to zero when temperature increases, suggesting a long-range ionic migration. This process may involve MA cations and I anions, as well as thermally created vacancies and point defects [60].

Regions I and II in $M^{\prime }\left(\nu \right)$ are marked by multiple well-defined peaks in $M^{\prime \prime }\left(\nu \right)$ (Figure 10b), suggesting the existence of different conduction mechanisms in certain frequency ranges. The shifting of the $M^{\prime \prime }\left(\nu \right)$ peaks toward higher frequencies with increasing the temperature indicates thermally activated dielectric relaxation processes [61]. The interfacial polarization aroused from ionic transport [62,63] and charges separation [56,64] at boundaries are dominating processes of different length scales, strongly depending on temperature.

A Gaussian fit was used to define the peaks’ position (Figure 10c) used further to calculate the activation energies and relaxation times according to the Arrhenius plots depicted in Figure 10d. For each of the frequency Regions I and II, two slopes are determined from Arrhenius plots, the change in slope taking place around 60 °C. Below this temperature, the activation energies are approx. 0.7 eV and 0.48 eV for Region I and Region II, respectively. In addition, Region I is characterized by approximately three orders of magnitude faster dynamics (relaxation time of ~10^−14^ s), compared to Region II (relaxation time of ~10^−11^ s). The differences between Regions I and II reduce for temperatures above 60 °C, suggesting that similar species are involved in the relaxation process above this temperature. According to a combined experimentally and theoretical study performed by Bakulin et al. [65] on MAPI films, at ambient temperatures the MA^+^ dipoles reorient on two distinct time scales of molecular motion, fast (of fs) and slow (of ps), directly affecting the dielectric response while at 180 °C the data can be fitted with a single time scale of ~6 × 10^−13^ s. These findings can be associated with the data obtained on the two distinct regions, Region I and Region II, revealed in our study. Thus, in our opinion, below 60 °C Region II accounts for the average dielectric response related to local MA^+^ molecular dipoles reorientations with respect to the iodide lattice, resulting in conductivity-related space-charge mobility in the crystal bulk at short times scales while Region I accounts for the stronger MA^+^ interfacial and boundary interactions giving rise to relaxation phenomena at long time scales. The average response in both Regions I and II measured between 60 °C and 110 °C exhibit similar dynamic processes on a 10^−13^ ÷ 10^−12^ s time scale. A significant change at ~60 °C is also observed in the Arrhenius plot of Range I where the activation energy change from 0.69 eV below 60 °C to 0.56 eV above (see Figure 10d). Similar behavior was reported for MAPI thin films in a frequency range similar to our Range I at ~45 °C when ion activation energy was measured to change from 0.7 eV to 0.5 eV, and it was related to a variation in crystal symmetry [66].

#### 3.3.3. Electrical Conductivity Properties

The AC conductivity is derivable from complex quantities of dielectric permittivity according to [33]
(6)$$\sigma_{ac}=\omega \varepsilon_{0}\varepsilon^{\prime \prime }\left(\omega \right),\;$$
with $\varepsilon^{\prime \prime }\left(\omega \right)=\varepsilon^{\prime }\left(\omega \right)\tan\;\delta =\frac{\sigma_{dc}}{\omega \varepsilon_{0}}$, ($\varepsilon_{0}=8.854\times 10^{-12}AsV^{-1}m^{-1}$) in which $\sigma_{dc}$ is the d.c. conductivity of the sample, and $\varepsilon_{0}$ is the dielectric permittivity of vacuum.

The temperature dependence of the complex permittivity components $\varepsilon^{\prime }$, $\varepsilon^{\prime \prime }$ and loss tangent (tan δ) at low frequencies, between 0.01 Hz ÷ 10 Hz, is illustrated in Figure 11. Dielectric permittivity reaches values of hundreds near 10 Hz and of thousands for 0.01 Hz, being accompanied by a spectacular growth of the dielectric losses that reach a plateau of few tens above 50 °C.

The increase in the permittivity at low frequencies most likely originates from the migration process that appears between the two main compounds of MAPI, MA, and PbI_6_ [67], resulting in a mixed ionic–electronic conduction. Unlike the oxide perovskites, where the most common migrating species are single or double ionized oxygen vacancies, the crystalline structure of MAPbI_3_ favors the migration of I^2−^, Pb^2+^ and CH_3_NH_3_^+^ ions. Additionally, the small formation energy necessary for I^−^ vacancies [68] leads to the highest probability of occurrence and concentration among all vacancies’ species. Although the concentration of vacancies is in balance at RT, the migration process is thermally activated at elevated temperatures favoring the diffusion of vacancies.

The amplified effect in the loss factor suggests a considerable ionic conductivity, originating from migration processes. By increasing the temperature, the molecular movement plays a key role in space charge polarization and charge/vacation hopping, resulting in an increase of $\varepsilon^{\prime }\left(T\right)$ with maxima in $\varepsilon^{\prime \prime }\left(T\right)$ and $\tan\;\delta \left(T\right)$. In the proximity of transition, such processes amplify.

The relaxation phenomena evidenced in the frequency dependence of the complex modulus should also be found in the frequency-dependent conductivity represented in Figure 12, where the thermally activated migration process totalizes as an overall effect.

The relaxation phenomena, with their widespread action in the frequency Regions I and II separated in the complex modulus formalism, are retrieved as an effective overall relaxation process in the 10^0^ Hz–10^4^ Hz range of *σ_ac_*(*ν*). This is shown in Figure 12a and analyzed in Figure 12b. Similar relaxation processes observed in *σ_ac_*(*ν*) were ascribed to ions and space charges hopping between adjacent vacancies at low and medium frequencies [60,69]. Based on DFT calculation, Frost et al. [10] assumed that the dipole–dipole interactions between the highly rotationally mobile MA^+^ form large ordered domains, which are responding slowly to the frequency of the applied electric field. When crossing the perovskite or accumulate at the interfaces the migrating species follow the applied field resulting in a noticeable long-range ionic conduction at low frequencies. The frequency-independent $\sigma_{dc}$ conductivity is analyzed in Figure 12c. The spatial charges contributing to the long-range ionic conductivity have activation energies of ~0.85 eV and 0.62 eV below and above 60 °C, respectively (Figure 12c). An increase in $\sigma_{dc}$ from ~10^−12^ Sxcm^−1^ up to ~10^−7^ S⋅cm^−1^ is measured between −30 °C and 110 °C. Worth mentioning is that the evidence of a phase transition is unclearly marked by the specific Curie point in the temperature dependence of permittivity. However, the Arrhenius analyses of both $M^{\prime \prime }\left(\nu \right)$ and $\sigma_{ac}\left(\nu \right)$ allows identification of regions described by two different slopes with a crossing point at 60 °C, which could be associated in our opinion with the tetragonal-to-cubic structural phase transition, preserving alike behavior while heating and cooling. This statement is also supported by structural investigations (Figure 2).

In the following, we will focus on the activation energies determined for the relaxation phenomena observed in $M^{\prime \prime }\left(\nu \right)$ (Figure 10d), $\sigma_{ac}\left(\nu \right)$ (Figure 12b) and $\sigma_{dc}\left(T\right)$ (Figure 12c). Different theoretical approaches have shown activation energies that have been attributed either to different charge carriers likely to migrate under certain conditions or to their ionized vacancies. In MAPI, thermally activated processes disclosed by distinct activation energies were assigned to vacancy-mediated migration of I^−^ and MA^+^ [24,70], which are the main category predisposed to migrate or diffuse in the lattice. Activation energies ranged between 0.08 eV to 0.6 eV were reported for iodine ions and their polarized vacancies [37,67,68,70,71,72,73,74,75,76,77], concluding that the iodine defects contribute to the diffusion processes in MAPI. Activation energies ranged between 0.43 eV and ~0.9 eV and were determined for MA^+^ ions and the corresponding polarized vacancies [67,68,69,70,71,72,75,76,77,78,79], pointing out to the MA^+^ ionic migration activation energy, which may differ depending on the cubic or tetragonal phase. Worth mentioning is that although numerous studies publish activation energies resulted in the migration of lead ions, other theoretically studies suggest that Pb ions are less likely to migrate [67,77], even if they can generate electrical polarization by displacement in the PbI_6_ octahedron [10]. Activation energies of ~0.80 eV [71] and of ~2.31 eV [67] were reported for polarized Pb vacancies.

We obtained activation energies of ~0.69 eV/0.48 eV at low temperatures, and of ~0.56 eV/0.5 eV at higher temperatures in two distinct frequency regions observed in $M^{\prime \prime }\left(\nu \right)$.

In $\sigma_{ac}\left(\nu \right)$ an overall relaxation phenomenon is observed on a wide frequency range characterized by energy barriers of ~0.43 eV at low temperatures and 0.48 eV at higher temperatures, approaching the theoretical DFT calculations for interstitial MA^+^ ions [68] and experimental results for the cubic state [75,77]. From $\sigma_{dc}\left(T\right)$, activation energies of ~0.85 eV at low temperatures and ~0.62 eV at higher temperatures were determined. In consistence with DFT calculations [68] and experiments [69], the main contributors to the long-range ionic conduction, in this case, are the MA^−^ vacancies. The iodine ions imprint could not be highlighted, and therefore for this purpose, an analysis with a focus set on the complex impedances is performed.

#### 3.3.4. Impedance Analysis

There are several interrelated functions that can be used to emphasize a charge transfer process with a certain time constant that would respond in a specific frequency range. In order to effectively interpreting the overall electrical behavior of the charge and ionic transport in the MAPI crystal and to develop a proper equivalent circuit model, we chose to express the experimental data in terms of complex impedances. The magnitudes of the resistive (real part of the impedance, ReZ) and the reactive (imaginary part of the impedance, −ImZ) components will be further used in different representations to separate the prevailing events attributed to the grains (bulk), grain boundary, and interface polarization occurred in the MAPI crystal.

ReZ decreases with increasing the frequency and temperature (Appendix A) accounting for the negative temperature coefficient of the resistances (NTCR) in crystalline materials [80]. Therefore, a decline of the bulk and interfaces resistances when raising the temperature may be anticipated, being in agreement with the enhanced $\sigma_{ac}\left(\nu \right)$ electrical conductivity. The slope in the representation −ImZ vs. frequency (Appendix A) corresponds to an ideally capacitive blocking electrode (slope = −1) [81] and deviates from the −1 value when the temperature increases above 0 °C (becomes +0.66 at 110 °C), especially in the lower frequency decades. This change of the slope reflects the transition toward a distribution of processes assigned to the interfacial heterogeneity and/or charge transfer reactions characterized by a distribution of relaxation times. Based on these premises, the equivalent circuit will necessarily contain a constant phase element (CPE) in its structure to make it suitable for all experimental conditions

The complex impedance plane ReZ vs. −ImZ, known as the Nyquist plot, is depicted in Figure 13a–c. It indicates a strong activity in the medium and low-frequency regions, associated with boundaries and contact interfaces. The impedance data measured at −30 °C presents a high tail in the low-frequency region (Figure 13a) that is diminishing at higher temperatures. The circular arc hardly starts to form at 0 °C and a perfectly clear and closed semicircle whose tail attempts to give outline to a second semicircle at very low frequencies is observable at 50 °C (see Figure 13b). When temperature increases even more, the semicircle diameter together with the low-frequency tail drastically decreases (Figure 13b,c). This type of evolution with temperature clearly indicates a thermally activated charge transport. Overlapping semicircles in the Nyquist plots and the mentioned relaxations are likely in real crystals [82].

The circular arcs in the impedance spectra Figure 13a–c mark the electrical properties of the bulk and boundary interfaces. Charge’s mobility manifests as a Warburg-like diffusion feature through (1) a semi-infinite transfer line, accounting for diffusion–recombination of electrons [83] at high frequencies and (2) a semi-infinite diffusion length, which evolves with increasing of temperature to a finite layer thickness diffusion length [84] for frequencies below 100 Hz. The appropriate equivalent circuit for fitting the experimental data in the investigated temperature and frequency ranges must describe the different time constants associated with the processes in the bulk and at the boundaries and the exacerbated mass transfer of ionic diffusion in lead–halide hybrid perovskites.

Our proposed equivalent electric circuit used to fit the Nyquist plot (Figure 13a–c) and the Bode plots (see Appendix A) consists of resistors (R), capacitors (C), and constant phase element (CPE), accounting for crystal inhomogeneity [85,86,87], and an impedance element (Z_W_) shorted by a resistor, accounting for a finite length Warburg-like diffusion [66,88]. Among the possible equivalent circuits, the most appropriate one fully fitting all the frequency decades at any of the investigated temperatures is depicted in Figure 13d, and it comprises a series of three groups of R/C elements, accounting for the following:(i)The geometrical resistance/capacitance of the bulk perovskite grains (R_g_/C_g_) in the high-frequency range, with the corresponding τ_grains_ time constant;(ii)The grain boundaries resistance/capacitance (R_gb_/C_gb_ elements corrected by CPE) [82] in the mid-frequency range with the corresponding τ_bounderies_ time constant;(iii)The accumulation/diffusion of charges at low-frequency by considering the R_ct_ − Z_W_/C_dl_ group with the corresponding τ_interfaces_ time constant. R_ct_ accounts for the resistance of coupled electron/ion transport, C_dl_ is the capacitance associated with the charges accumulated or adsorbed at the dielectric–electrode interfaces of the opposing polarity. and the Warburg Z_W_ complex impedance accounts for the diffusion of the electroactive species [89]. We define three-time constants for each of the RC branches in the equivalent circuit in Figure 13d, τ_grains_, τ_boundaries_, and τ_interfaces_.

The temperature dependencies of all the R and C components are shown in Appendix A. The temperature dependence of the time constants is given in Figure 13e. The corresponding activation energies of charge transport associated with the grains resulted from linear fit on τ_grains_(*T*) are Ea_g1_ = 0.47 eV at low temperatures and Ea_g2_ = 0.35 eV at high temperatures. According to DFT calculations performed by Yang et al. [68], these values stand out for interstitial MA^+^ ions; however, our extracted energies are consistent with other lately theoretical calculations and experiments [70,71,72,75,77,78,79]. As it can be observed, τ_grains_ changes at about 60 °C. We connect this behavior with the phase transition occurring at this temperature, generating two different slopes in the temperature dependence of the time constant associated with the grains. The distinct activation energies for τ_grains_ indicate the manner in which the orientation of the dipole in the crystal interacts with the charge carriers when temperature increases, activating ions mobility and their diffusion. At temperatures above 60 °C, a longer time is necessary to charge C_g_, as opposed to the one below 60 °C.

The activation energy of charge transport associated with the grain boundaries extracted from linear fit on τ_bounderies_(*T*) (Ea_gb_ = 0.74 eV) approaches previously reported values [68,69,75,76] related to the polarized MA vacancies. The lowest activation energy resulted from τ_interfaces_(*T*) (Ea_i_ = 0.28eV) for the interfaces region, which is also very close to those previously reported for interstitials iodine ions and their vacations, based on both theory and experiments [37,68,72,73,74,75,76,77]. The temperature increasing reveals a shorter time necessary to charge C_gb_ in contrast to the one for C_dl_. Further, τ_interfaces_ > τ_boundaries_ at temperatures above 60 °C indicates a lower Debye length [90] and consequently a larger number of ions at the interfaces than at the grain boundaries.

## 4. Conclusions

The electrical, ferroelectric, dielectric, and relaxation properties of hybrid perovskite MAPI crystals are investigated in order to characterize the intrinsic electrical behavior of the material in the temperature range of interest for photovoltaic applications. Highlighting the fact that reliable information on intrinsic material properties requires studies conducted on single-crystal samples, the results obtained in this study are also useful for the refinement of theoretical calculations and simulation of MAPI-based devices, including perovskite solar cells.

The observed phenomena are explained in accordance with the structural aspects, establishing a connection between the nature of the migrating species and their impact on the macroscopic properties. At the same time, the results bring arguments for the existence of ferroelectricity.

The dense and compact MAPI crystals synthesized via inverse temperature crystallization contain large crystalline regions. PFM images highlighted ferroelectric domains at the surface of the sample.

The XRD studies, performed between 27 °C and 110 °C, evidenced that the phase transition from tetragonal to cubic occurs gradually, the two phases coexisting up to 60 °C, when the tetragonal lines apparently merge into one slightly asymmetric XRD peak. Increasing the temperature, the unit cell volume expands, and a large structural inhomogeneity is observed. The gradual phase transition is confirmed by the dielectric function and was analyzed in detail with respect to frequency and temperature.

The dielectric function exhibits a strongly diffusive character over the whole 10 mHz to 1 MHz frequency range, wherefrom originate the frequency dispersive relaxation phenomena with distinct time constants, each having its own influence on the thermally activated conduction mechanisms.

There are two relaxation phenomena that dominantly act in the medium- and low-frequency regions, producing a cumulative effect in the $\sigma_{ac}\left(\nu ,T\right)$. They were disjointed by analyzing the complex dielectric function combined with the dielectric modulus approach. Based on the extracted energy barriers, the two relaxation processes were identified as originating from the movement of interstitial MA^+^ ions, mediated by their polarized vacancies, with distinct responses to the frequency of the small AC applied electric field. Thus, the MA^+^ ions’ response is observed at medium frequencies and is associated with ions’ accumulation at the grain boundaries, while the negative polarized MA vacancies may cross the perovskite and accumulate at the interfaces with electrodes, resulting in long-range ionic conduction that is observed at low frequencies.

The contribution of the iodine ions to the diffusion processes in MAPI crystal was determined by developing an equivalent circuit model, fully describing the experimental data in the whole range of explored temperatures (−30 °C ÷ 110 °C) and frequencies (10^−2^ Hz ÷ 10^7^ Hz). Thus, the main regimes of ionic transport in the MAPI crystal were determined by analyzing the impedance spectra and dissociating between the frequency and temperature-dependent signals emanating from the bulk of crystal, boundaries, and interfaces, contributing each to the overall conductivity. The activation energies extracted from Arrhenius plots of the equivalent circuit time constants are found to be consistent with the lately DFT calculations and previous experiments. In this way, we conclude that the interstitial MA^+^ ions, together with their vacancies, produce thermally activated relaxation phenomena related to the grains and interfaces. The polarized vacancies are the main species liable to accumulate at the grain boundaries and interfaces involving distinctive thermally activated relaxation phenomena at long-time scales and conduction mechanisms. At the grain boundaries, the prevailing response is generated by the MA^+^ and their associated polarized vacancies, masking the iodine ions’ contribution in the $\sigma_{dc}\left(T\right)$. However, by analyzing the temperature dependence of the equivalent circuit time constant ascribed to the diffusion process at the interfaces, the imprint of the iodine ions was recognized based on its energy barrier of 0.28 eV.

The high reproducibility of the electrical data indicates good thermal stability of the material during its exposure to −30 °C–110 °C temperature variations, which is an important attribute for halide perovskites applications. In addition, the presented results provide insights into the correlation between the properties of the crystal and the interfacial dynamics. The increase of the intrinsic bulk ionic conductivity upon increasing temperature and the accumulation of mobile ions at interfaces happen in absence of light, due only to the application of an external small AC electric field.

## Figures and Tables

**Figure 1 materials-14-04215-f001:**
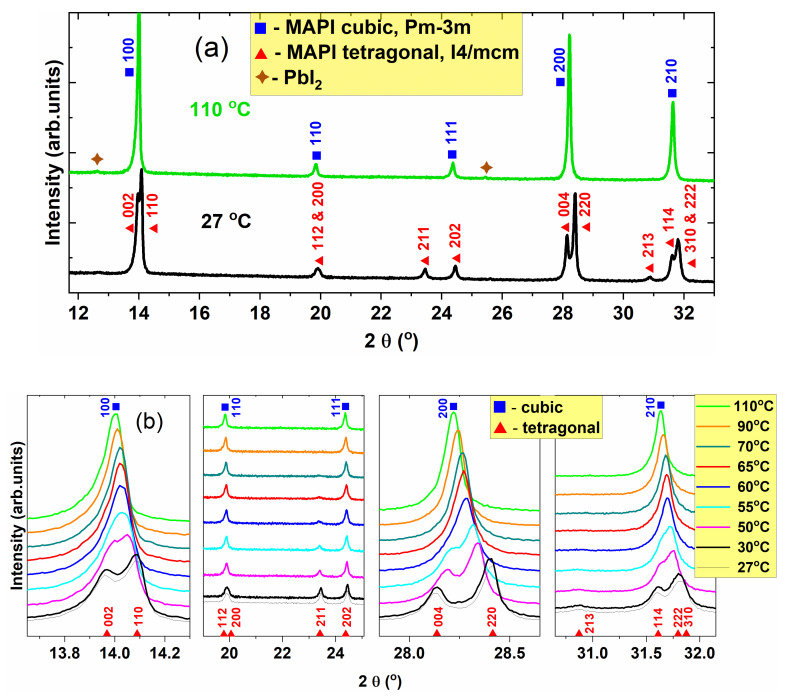
XRD patterns: (**a**) recorded at 27 °C and 110 °C, showing the phase identification and line indexing; (**b**) XRD at different temperatures, showing zoomed views of the significant regions. The diffractograms are represented after removing the contribution of CuKα_2_ radiation. The diffractograms were vertically shifted.

**Figure 2 materials-14-04215-f002:**
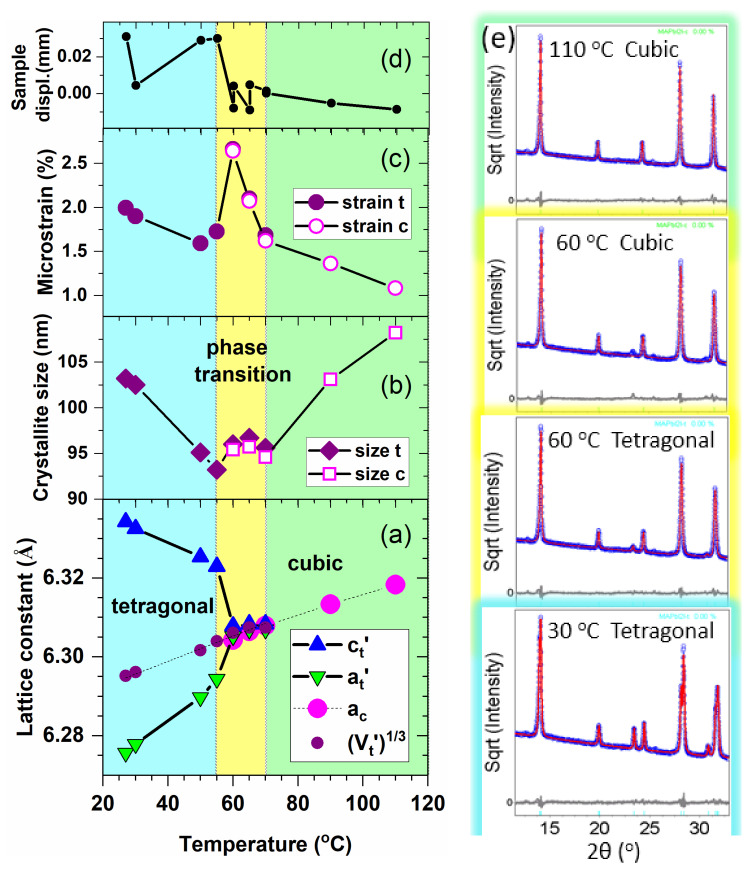
Evolution of the structure parameters determined by XRD: (**a**) lattice constants; (**b**) crystallite size; (**c**) microstrain; (**d**) sample displacement; (**e**) fitted diffractograms: experimental data (blue points), simulated profile (red line) and difference curve (grey line). Between 60 and 70 °C, the diffractograms were fitted either with a cubic or a tetragonal structure.

**Figure 3 materials-14-04215-f003:**
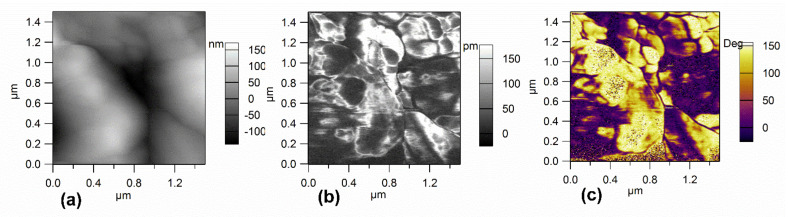
MAPI crystal: (**a**) AFM topography, (**b**) piezoresponse amplitude, and (**c**) piezoresponse phase.

**Figure 4 materials-14-04215-f004:**
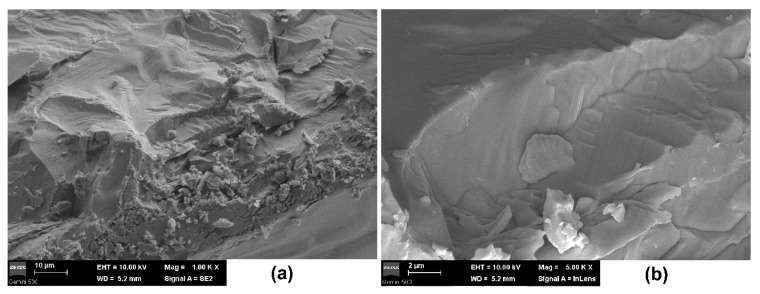
SEM images with different magnifications: (**a**) 10 µm; (**b**) 2 µm.

**Figure 5 materials-14-04215-f005:**
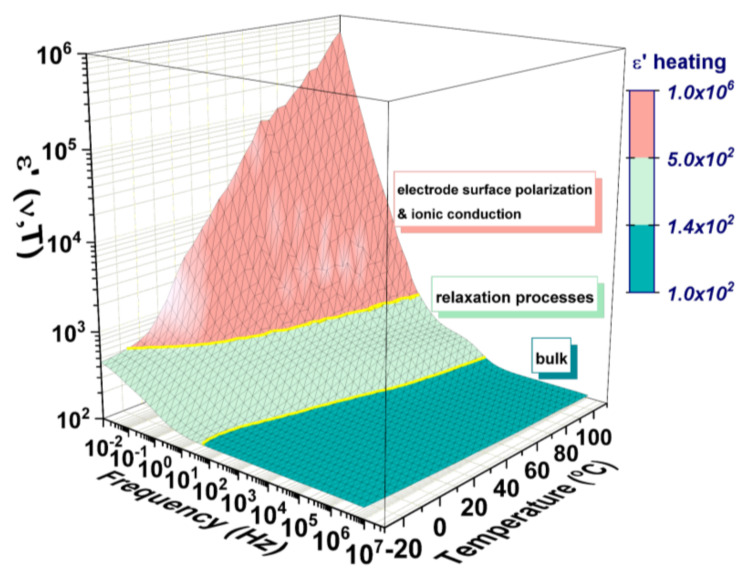
The real part of dielectric permittivity represented as a function of frequency and temperature at heating (for cooling see Appendix A).

**Figure 6 materials-14-04215-f006:**
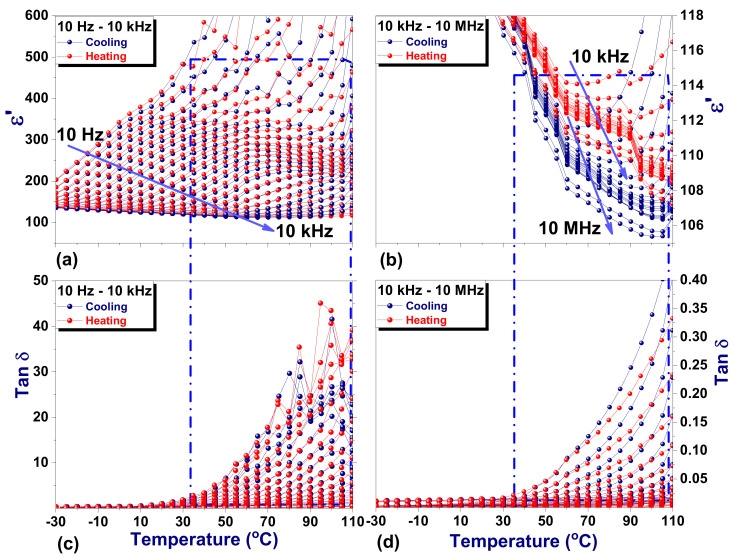
The temperature variation of ε’ at different frequency intervals: (**a**) 10 Hz–1 kHz; (**b**) 10 kHz–10 MHz and similarly tanδ at different frequency intervals: (**c**) 10 Hz–1 kHz; (**d**) 10 kHz–10 MHz.

**Figure 7 materials-14-04215-f007:**
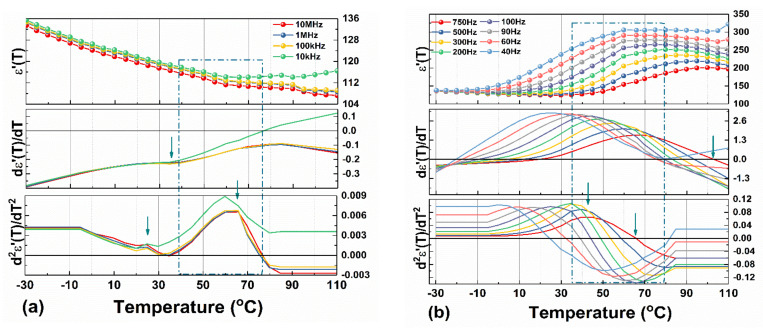
The real part of the dielectric constant measured during heating and the corresponding first- and second-order derivatives with respect to temperature for a few selected frequencies in the ranges: (**a**) 10 kHz ÷ 10 MHz; (**b**) 40 Hz ÷ 750 Hz.

**Figure 8 materials-14-04215-f008:**
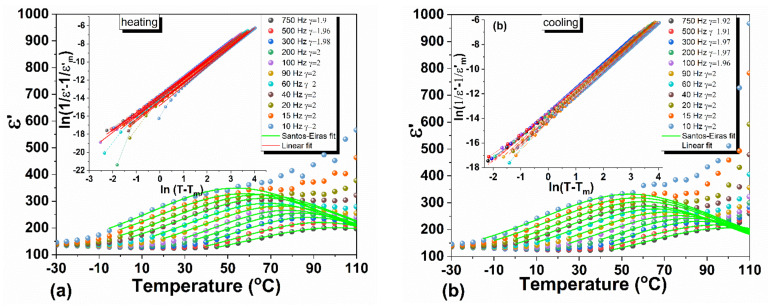
Temperature dependence of $\varepsilon^{\prime }\left(T\right)$ in the 10 Hz ÷ 750 Hz frequency range: (**a**) during heating and (**b**) cooling. The nonlinear fits are according to the modified Curie–Weiss law and the linearized plots are based on the generalized Curie–Weiss law (in the corresponding insets).

**Figure 9 materials-14-04215-f009:**
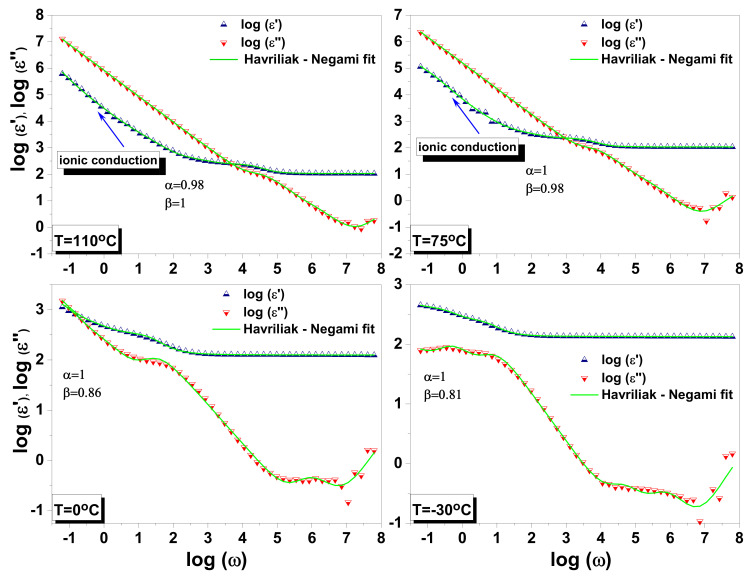
Complex permittivity components $\varepsilon^{\prime }\left(\omega \right)$ and $\varepsilon^{\prime \prime }\left(\omega \right)$ as a function of angular velocity (ω = 2πν) illustrated for a few selected temperatures.

**Figure 10 materials-14-04215-f010:**
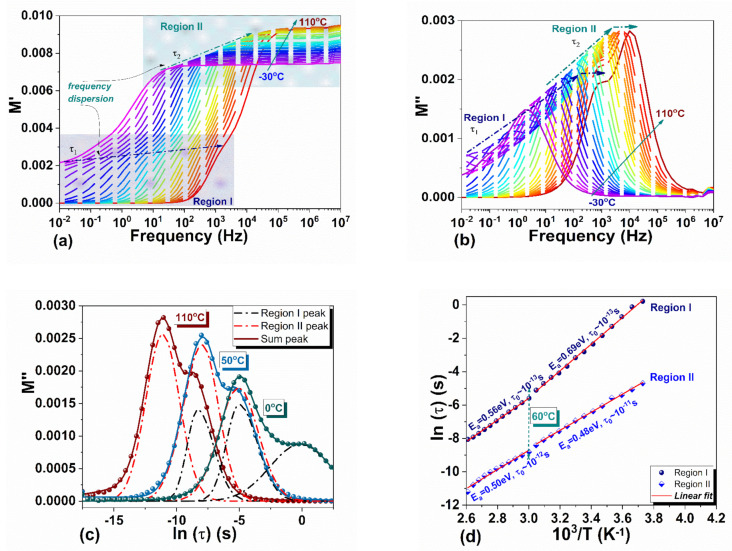
Complex modulus *M** and relaxation time between −30 °C and 110 °C: (**a**) $M^{\prime }\left(\nu \right)$; (**b**) $M^{\prime \prime }\left(\nu \right)$; (**c**) Gaussian fit of the maxima in *M*″; (**d**) relaxation time during cooling.

**Figure 11 materials-14-04215-f011:**
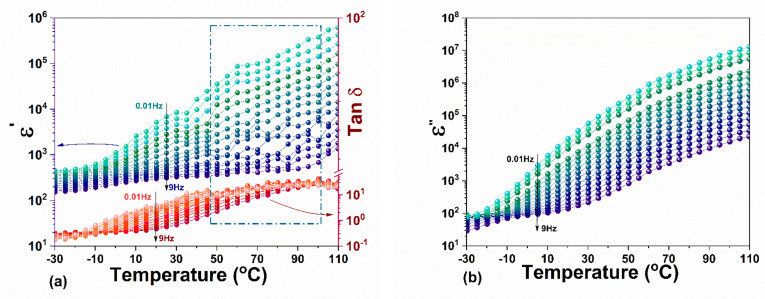
The temperature dependence of (**a**) the dielectric permittivity ε′ and loss tangent tan δ and (**b**) the loss factor ε″ measured during heating and cooling in the frequency ranges of 0.01 Hz ÷ 10 Hz (for the full frequency range up to 10 MHz see Appendix A).

**Figure 12 materials-14-04215-f012:**
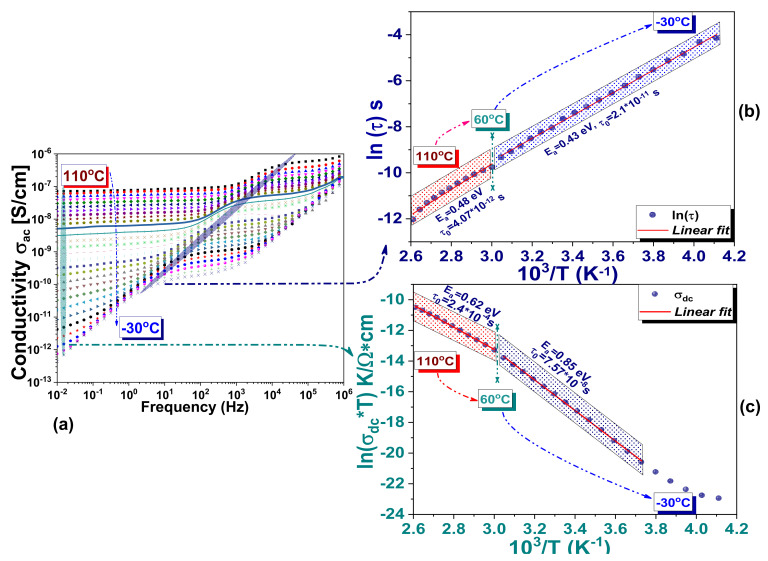
(**a**) Variation of ac conductivity as a function of frequency *σ_ac_*(*ν*) measured between −30 °C and 110 °C; (**b**) linear extrapolation following the relaxation in frequency dependence of the a.c. conductivity in the 10^0^ Hz ÷ 10^4^ Hz range; (**c**) Arrhenius plots for d.c. conductivity in the low-frequency range.

**Figure 13 materials-14-04215-f013:**
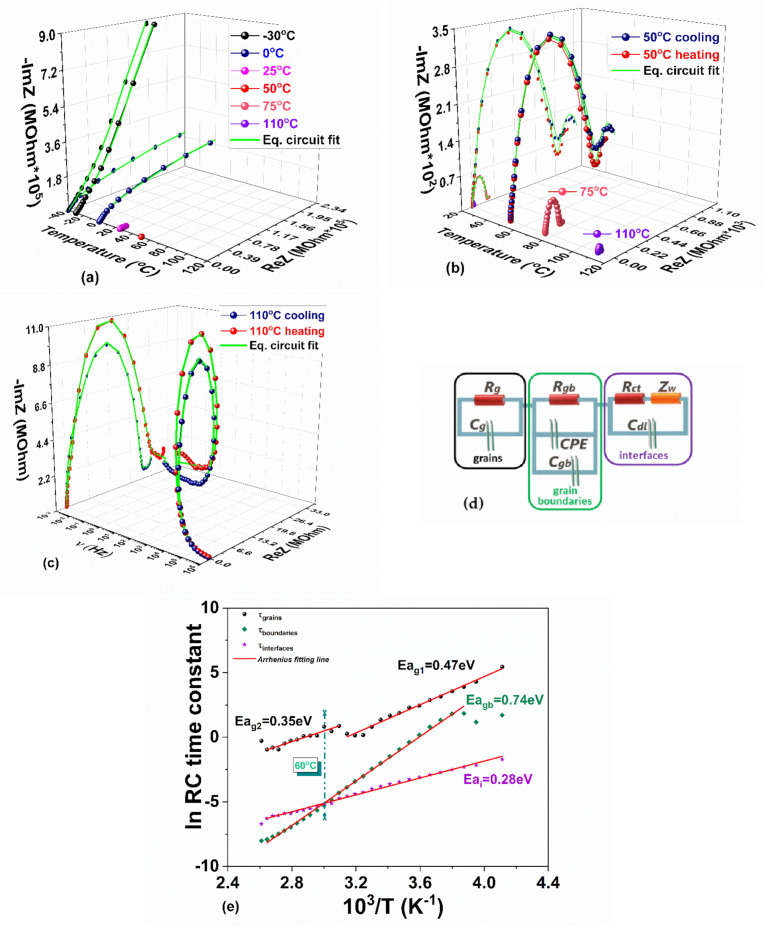
(**a**) Nyquist spectra for a few selected temperatures in the range of −30 °C and 110 °C together with (**b**) zoom for higher temperatures and (**c**) zoom for 110 °C for both paths of heating and cooling; (**d**) the equivalent circuit used for fitting the Nyquist spectra; (**e**) Arrhenius-like activation energies of the time constant related to grain, grains boundary and interfaces.

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
