# Peer review of "Tetragonal–Cubic Phase Transition and Low-Field Dielectric Properties of CH3NH3PbI3 Crystals"

_materials, 2021, doi:10.3390/ma14154215_

Round 1
Reviewer 1 Report
MAPI crystals are of interest to researchers for their ferroelectric properties, piezoelectric and electrostrictive characteristics. In this work, the dielectric properties of the MAPI crystal were studied as a function of the temperature and the frequency of the external electric field. The authors used a set of methods for studying microstructure, surface morphology, dielectric spectroscopy, analysis of relaxation of dielectric properties, and impedance studies. A very large amount of work has been done.Конец формы
Studies have made it possible to establish the electrical properties of hybrid perovskite crystals, factors affecting the dielectric constant and electrical conductivity in a wide temperature range. It is shown that an increase in temperature leads to a smooth phase transition from a tetragonal structure to a cubic one. The presence of two mechanisms of dielectric relaxation in different frequency ranges has been established and explained. There is no doubt about the reliability of the results obtained in the work, the work leaves a favorable impression. The results of these studies are in demand, because serve to accumulate knowledge about materials that are promising for practical use.
Reviewer 2 Report
The presented manuscript by Roxana E. Patru and co-authors reports the synthesis and detailed electrical characterization of widely investigated CH3NH3PbI3. In my opinion the paper is worth to be published in Materials, however, according to the Journal standards the minor revision is required. Below I listed some modifications to improve the paper before publication.
- The quality of presented Figures 1, 2, 5, 13 are of poor quality.
- I believe that the discussion on ferroelectricity is exaggerated, where observing different areas on the PFM is not a rationale for ferroelectricity.
- It is difficult to see the consistency between the Fig 6 a and b. Why the data from one frequency decade were omitted?
- Data analysis using first and second derivatives using such measurement resolution is debatable.
- Why was the conductivity part not taken into account during the fitting procedure (Formula 4)? Moreover, in my opinion, the paragraph describing the various relaxation models is only a “filler” and should be omitted.
Reviewer 3 Report
MAPI crystals have recently attracted a lot of attention due to their ferroelectric, piezoelectric and electro-strictive properties. This manuscript contains a detailed experimental analysis of hybrid perovskite MAPI crystals. The authors have conducted a very large amount of experimental studies to study microstructure, surface morphology, dielectric spectroscopy, relaxation analysis of dielectric properties, and impedance studies. The advantage of this study is that it was able to establish the electrical properties of hybrid perovskite crystals, which are factors affecting the dielectric constant and electrical conductivity over a wide temperature range. The results of these studies are valuable in that they accumulate knowledge about materials with potential for practical use. I have a few concerns as follows, and I hope the manuscript will be improved accordingly.
In lines 74 and 690, “-30 °C ÷ +110 °C”. Double-check the symbols in the middle and correct them if necessary.
The resolution in Figures 1 - 5 is too low, so the figures are barely readable. They need to be improved for the high-quality publication in Materials.
In order to highlight the value of the study, it would be good if the practical importance of MAPI crystals was emphasized in the conclusion, respectively.
Reviewer 4 Report
The authors present the frequency and temperature dependent dielectric properties of CH3NH3PbI3 and try to relate these to the structural phase transition and some other microstructure information, aiming at discovering the conduction mechanisms and other underlying physics. Overall, the manuscription can be accepted for publication. Some suggestions could be as follows:
Between 30°C and + 110°C, not necessary to use + here.
The resolution of the figures is too low, Hard to see the text in the figures.
In Fig. 1a, 110°C instead of 110°
In Fig. 2b, it is unclear for ‘crysTable 70. °C’. there are no descriptions from c to e in the caption.
Some details on calculating the crystallite size and the microstrain.
How did you prepare the nanostructured materials from 3-5mm sized crystals?
At high temperatures, because of the large atomic displacement parameters, the peak width could be enlarged.
For good comparison, it better to use uniform unit for the temperature (°C or K)
